# ViT-AE++: Improving Vision Transformer Autoencoder for Self-supervised Medical Image Representations

**Chinmay Prabhakar**[*1]                                          CHINMAY.PRABHAKAR@UZH.CH

**Hongwei Bran Li**[*1,2]                                          HONGWEI.LI@TUM.DE

**Jiancheng Yang**[3]                                          JIANCHENG.YANG@EPFL.CH

**Suprosanna Shit**[2]                                          SUPROSANNA.SHIT@TUM.DE

**Benedikt Wiestler** [†4]                                          B.WIESTLER@TUM.DE

**Bjoern H. Menze** [†1]                                          BJOERN.MENZE@UZH.CH

[1] *Department of Quantitative Biomedicine, University of Zurich, Switzerland*

[2] *Department of Computer Science, Technical University of Munich, Germany*

[3] *Computer Vision Laboratory, EPFL, Switzerland*

[4] *Klinikum rechts der Isar, Technical University of Munich, Germany*

**Editors:** Accepted for publication at MIDL 2023

## Abstract

Self-supervised learning has attracted increasing attention as it learns data-driven representation from data without annotations. Vision transformer-based autoencoder (*ViT-AE*) (He et al., 2021) is a recent self-supervised learning technique that employs a patch-masking strategy to learn a meaningful latent space. In this paper, we focus on improving *ViT-AE* (nicknamed *ViT-AE++*) for a more effective representation of both 2D and 3D medical images. We propose two new loss functions to enhance the representation during the training stage. The first loss term aims to improve self-reconstruction by considering the structured dependencies and hence *indirectly* improving the representation. The second loss term leverages contrastive loss to *directly* optimize the representation from two randomly masked views. As an independent contribution, we extended *ViT-AE++* to a 3D fashion for volumetric medical images. We extensively evaluate *ViT-AE++* on both natural images and medical images, demonstrating consistent improvement over vanilla *ViT-AE* and its superiority over other contrastive learning approaches. Our code is available at https://github.com/chinmay5/vit_ae_plus_plus.git

**Keywords:** representation; self-supervised learning; masked vision transformer

## 1. Introduction

Self-supervised representation learning (SSRL), especially recent contrastive learning-based methods (Chen et al., 2020; He et al., 2020; Oord et al., 2018), is a promising technique that can learn informative data representations from data without labels. It is particularly useful and relevant in medical imaging, where labeled data are often scarce for traditional supervised training and the high cost of manual labeling that relies on domain knowledge.

Contrastive learning-based approaches aim to attract *similar* image pairs and rebuff *dissimilar* image pairs. A similar image pair can be generated by two augmented views of

---

[*] Contributed equally

[†] Equal senior supervision

one image, and the other image samples can be used to construct dissimilar image pairs. In medical imaging, *SSRL* are mainly used for two purposes: 1) pre-training deep networks for transfer learning or for network initialization (Zhou et al., 2019; Chaitanya et al., 2020; Zeng et al., 2021), 2) extracting meaningful information from data (Li et al., 2021; Dufumier et al., 2021) which is the downstream task for applying *SSRL* in this work.

As opposed to contrastive learning, the recent vision transformer autoencoder (*ViT-AE*) approach (He et al., 2021) is different from the above methods in principle. It randomly masks the sequential image patches and learns to reconstruct the original image using an autoencoder and vision transformer-based architecture. They demonstrate that such a simple reconstruction can facilitate the vision transformer to learn effective image representations in a self-supervised fashion. Although *ViT-AE* achieves promising results in the natural image domain, we argue that two components of this framework could be further optimized.

First, in the vanilla *ViT-AE* pipeline, it computes pixel-wise reconstruction loss of the autoencoder and does not consider structured dependencies of the reconstruction, which might limit to capture of semantic features. For example, medical images contain rich texture and morphology, such structural information across pixels is important to complement traditional pixel-wise reconstruction (often with a $\mathcal{L}$-2 norm loss).

Second, since the representation is *indirectly* learned by a self-reconstruction via an autoencoder, there might be room to optimize the target representation directly. For example, contrastive learning-based methods (Chen and He, 2021; Chen et al., 2020) straightforwardly match the target representation from two augmented views.

**Contributions.** (1) We introduce an auxiliary reconstruction task that considers structural dependencies to complement the pixel-wise reconstruction. (2) We unite two paradigms of contrastive learning-based and autoencoder-based methods and enjoy the benefits of both. (3) In extensive experiments, we demonstrate that both 2D and 3D *ViT-AE++* outperform the vanilla *ViT-AE* and its superiority over other contrastive learning approaches, setting up a strong baseline for learning self-supervised medical image representation.

## 2. Related Work

We mainly discuss prior works that are related to the technical contributions of *ViT-AE++*.

**Auxiliary loss functions.** In image segmentation tasks, auxiliary losses can regularize network training by serving as 'deep supervision' (Lee et al., 2015; Zhao et al., 2017). Liu et al. (2021) introduce a generic perceptual loss for dense prediction tasks. They argue that leveraging an auxiliary task that considers structural dependency can benefit various dense prediction tasks. Inspired by this work, we introduce an auxiliary reconstruction task to the autoencoder. In addition to perceptual loss, we develop a new edge-aware loss considering the richness of texture in medical images.

**Contrastive loss.** Contrastive loss, such as *SimCLR* (Chen et al., 2020) directly optimizes the image representation by maximizing the agreement between two augmented views. The asymmetric networks such as *SimSiam* (Chen and He, 2020) and *BYOL* (Grill et al., 2020) follow a similar idea but only make use of positive pairs with two shared encoders. Considering the training efficiency, we borrow the loss design from *SimSiam* (Chen and

He, 2020). Differently, we not only use random augmentation but also random masking to obtain hard training samples, analogous to random cropping in *CNN*-based backbone.

## 3. Method

**Overview.** The objective is to learn a good domain-specific representation of 3D volumes using an autoencoder without labels. Consider an autoencoder with encoder $E$, decoder $D$, and a function $m(\cdot)$ for masking input patches in a vision transformer architecture. Given an image volume $X$, it is decomposed into $k$ sequential patches with a size of $n \times n \times n$. Then a random mask is applied on the sequential input patches to mask away $p\%$ of the patches. The remaining visible patches are referred to as $X^*$. The visible patches $X^*$ are fed to the encoder $E$ to extract features. For each missing patch, a token with 3D positional encoding is assigned to indicate the presence of such patches (called *MASK* tokens). The features of the visible patches along with *MASK* tokens are passed to the decoder. The decoder $G$ predicts image intensities for these *MASK* tokens and thereby reconstructs the whole volume, i.e., $X \approx D(E(X^*))$. We introduce an auxiliary reconstruction task with a compound loss function $\mathcal{L}_{per} + \mathcal{L}_{edge}$ as shown in Fig. 1. The new loss is designed to capture high-order properties to complement the pixel-wise loss. To further enhance the target representation, we adopt contrastive loss to maximize the agreement from two random masked views. Fig. 1 shows the schematic view of our architecture. In the following sections, we explain each component in detail.

**Vanilla *ViT-AE* with pixel-wise loss.** *ViT-AE* takes partial observations and reconstructs the original input. A random masking strategy is used to mask $p\%$ of the input volume. Notably, $p$ is a hyperparameter which will be discussed in a later section. The visible patches $X^*$ are fed through encoder $E$. The decoder $D$ reconstructs $X$ from $E(X^*)$ using a mean-squared-error loss: $\mathcal{L}_{rec} = ||X - D(E(X^*))||_2$.

**Auxiliary compound loss.** Original *ViT-AE* uses the pixel-wise loss for training as mentioned above. In the medical image domain, perceptual features and edges encode meaningful semantics signals (Shen et al., 2017). We employ a compound loss with deep high-level features and low-level edge-based features to enforce the network to use this supervision signal during training.

To exact deep multi-level features, we introduce *VGG*-based perceptual loss (Johnson et al., 2016) to compute the feature similarity at multiple levels over $n$ 2D slices of one volume.

$$\mathcal{L}_{per} = \sum_{i=0}^{n} ||VGG(x_i) - VGG(\hat{x}_i)||_2 \tag{1}$$

where $VGG(\cdot)$ denotes multi-level features from the pre-trained *VGG* network using the same layers in (Johnson et al., 2016). $x_i$ and $\hat{x}_i$ denote the input image and the reconstructed image, respectively.

For low-level features, a 3D *Sobel* filter function (Kanopoulos et al., 1988) $Sobel(\cdot)$ is used to compute the gradients from three directions. Firstly, the input volume is convolved with a fixed *Sobel* filter to generate gradients in the axial, coronal, and sagittal directions.

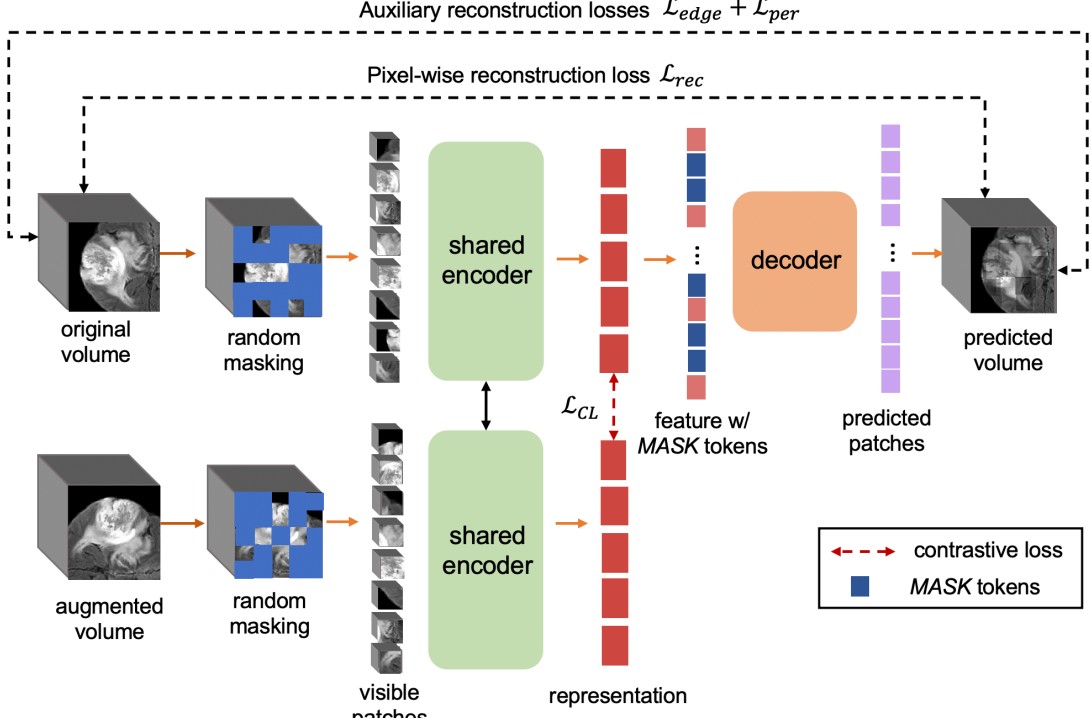

Figure 1: Our proposed *ViT-AE++* framework. The upper half shows the training procedure of the autoencoder. An MRI volume is parsed and masked randomly and the visible patches are passed through to the encoder. Placeholder tokens for the masked patches are introduced *after* the encoder (referred to as the *MASK* tokens). The full set of encoded features and these placeholder tokens are passed to a decoder which reconstructs the whole MRI volume. We use a pixel-wise reconstruction loss and an auxiliary loss that concerns perceptual features (Eq. 1) and edge map (Eq. 2). The lower half demonstrates the optimization with a contrastive loss which is based on two randomly augmented and masked views. The representations from the two views are matched with cosine similarity loss (Eq. 3).

The norm of these gradients is the edge map representation (Eq. 2). Please see the details in the Appendix.

The edge loss between original volume $X$ and the reconstruction $\hat{X}$ is formulated as:

$$\mathcal{L}_{edge} = ||Sobel(X) - Sobel(\hat{X})||_2 \tag{2}$$

**Contrastive loss.** Along with the self-reconstruction, a contrastive loss is further introduced to enhance the target representation. It tries to match the feature representations between the two views of visible patches, denoted as $f_1$ and $f_2$. We used negative cosine distance as the loss function and the 'stopping gradient' optimization in (Chen and He, 2020).

$$\mathcal{L}_{CL} = -\frac{f_1}{||f_1||_2} \cdot \frac{f_2}{||f_2||_2}, \tag{3}$$

**ViT-AE++.** The final optimization objective of *ViT-AE++* is

$$L = \mathcal{L}_{rec} + \lambda_1 \mathcal{L}_{per} + \lambda_2 \mathcal{L}_{edge} + \mathcal{L}_{CL}. \tag{4}$$

where $\lambda_1$ and $\lambda_2$ are hyperparameters and initially set to 0.01 and 10 respectively considering the loss scale. $\lambda_2$ is decreased linearly to stabilize the training process. The motivation for the linear scaling and other hyperparameters are discussed in the Results section.

## 4. Experiment Setup

We evaluate *ViT-AE++* extensively on public natural and medical image datasets, focusing on 3D brain MRI datasets.

**2D Datasets.** *CIFAR-10*, *CIFAR-100* and Tiny Imagenet-100 (Russakovsky et al., 2015) are popular natural datasets to benchmark representation learning methods. In addition, we evaluate our method on a 2D chest X-ray dataset (Kermany et al., 2018). It consists of 5118 images for training, 119 for validation, and 626 for testing. The task is to identify pneumonia based on chest X-ray images. All images are resized to 224×224 pixels.

**3D Datasets.** To highlight the effectiveness of 3D *ViT-AE++*, we performed experiments on two public 3D datasets: 1) a multi-center MRI dataset (*BraTS*) (Menze et al., 2014; Bakas et al., 2018) including 326 patients with a brain tumor. 2) The Erasmus Glioma Database (*EGD*) (van der Voort et al., 2021) which consists of 768 MRI scans. Both datasets include FLAIR, T1, T2, and T1-c with a uniform voxel size $1{\times}1{\times}1$ $mm^3$. We train individual models and evaluate on the two datasets separately. The effectiveness of the learned representations is evaluated on two down-stream classification tasks: a) For *BraTS*, discriminating high-grade and low-grade tumor, and b) for *EGD*, predicting isocitrate dehydrogenase (IDH) mutation status (0 or 1).

We use the segmentation masks for both datasets to get its centroid and generate a 3D bounding box of 96×96×96 to localize the tumor. If the bounding box exceeds the original volume, the out-of-box region is padded with background intensity. Four modalities are concatenated to serve as a multi-modal input. The intensity range of all image volumes was rescaled to [0, 255] to guarantee the success of intensity-based augmentations.

**Architecture and training configuration.** In this sub-section, we explain each component in the architecture and the training settings in detail.

*Encoder and decoder.* Working on only the visible patches allows the encoder to process a fraction of 3D volumes with less GPU memory. We extend a standard ViT (Dosovitskiy et al., 2021) to be 3D architecture as the encoder. 3D patches are embedded using a linear projection with added positional encoding with 3D coordinates. 12 encoder blocks, each with an embedding dimension of 768, are used. The decoder reconstructs the volume using features from the encoder and the *MASK* tokens mentioned in Sec. 3. Each *MASK* token is a shared learned vector that indicates the presence of a missing patch to be predicted. We add 3D positional encoding to all the tokens for sequential prediction (Devlin et al., 2019). The decoder is realized using 8 transformer blocks, each with an embedding dimension of 512. After training, we only use the encoder as a feature extractor. The 3D position encoding is shown in the Appendix.

*End-to-end training of* ViT-AE++ *with contrastive loss.* *ViT-AE++* is trained for 1000 epochs using *AdamW* optimizer (Loshchilov and Hutter, 2019) with 0.05 weight decay. The

base learning rate is 1e-3. The learning rate is annealed using cosine decay (Loshchilov and Hutter, 2017). The batch size is set to 4, adjusted to maximize GPU memory with an Nvidia RTX 3090. We use 40 epochs for our warm-up schedule (Goyal et al., 2018). *Gamma correction, affine transform and Gaussian noise* augmentations are used to generate the second "view" of the input volume. The two views are randomly masked and passed through the shared encoder $E$. Features generated by the encoder are compared using their cosine similarity (Eq. 3).

**Evaluation strategy, classifier, and metrics.** For the *EGD* dataset, since there are 307 unlabeled images, we pre-train on the unlabeled data and perform the downstream task on the labeled dataset. For the other 2D and 3D datasets, the training of the proxy task and the target downstream task use the same dataset.

For evaluation on *BraTS* and *EGD* datasets, we follow the standard strategy to evaluate the quality of the pre-trained representations by training a *supervised* linear support vector machine classifier on the training set and then evaluating it on the test set. We use the sensitivity, specificity and Area Under the Receiver Operating Characteristic Curve (AUC) as the evaluation metrics. We use *stratified five-fold nested cross-validation* to reduce selection bias and validate each model. In each fold, we randomly sample 80% subjects from each class as the training set and the remaining 20% for each class as the test set. Furthermore, 20% of training data is separated and used exclusively to optimize the hyper-parameters within each fold.

For evaluation on *CIFAR-10*, *CIFAR-100*, Tiny Imagenet-100, and chest X-ray datasets, we use the linear probing strategy (He et al., 2021). The decoder module is discarded and encoder weights are frozen. Thus, the encoder acts as a feature extractor. A supervised linear classifier is trained on the frozen representations. We use the pre-defined train and test splits. 20 % of the training data is separated for hyper-parameter tuning. We report the classification accuracy for the test split for all the datasets.

## 5. Results

**ViT-AE++ *vs*. ViT-AE on 2D datasets.** To quantify the effectiveness of our proposed loss functions, we compare our method against the vanilla ViT-AE on *CIFAR-10*, *CIFAR-100*, Tiny Imagenet-100 and *chest X-ray* datasets. We use the linear probing evaluation protocol and report the classification accuracy on the test sets. Please note that the VGG-perceptual loss for three natural image datasets is not a pre-trained VGG but one with randomized weights. This is because we do not wish any supervised signals involved during the self-supervised training stage. Tab. 1 shows that the features learned using ViT-AE++ consistently perform better than their vanilla counterpart. It indicates that ViT-AE++ can serve as a strong baseline for both natural and medical image domains.

Table 1: Comparison of ViT-AE++ and ViT-AE on four 2D datasets. We can observe consistent improvements in classification accuracy, benefiting from the new loss functions.

| Method | CIFAR-10 | CIFAR-100 | Tiny ImageNet-100 | chest X-ray |
|---|---|---|---|---|
| ViT-AE | 94.10 | 75.61 | 70.42 | 95.20 |
| ViT-AE++ (ours) | **95.40** | **78.82** | **72.09** | **95.60** |

**ViT-AE++ *vs.* other methods on 3D datasets.** To focus on 3D medical image representation on small-scale datasets, we further evaluate the 3D version of ViT-AE++ and compare it with other state-of-the-art self-supervised methods (especially contrastive learning-based ones) on two classification tasks: a) discrimination of low-grade and high-grade brain tumors and b) prediction of IDH mutation status.

We observe that ViT-AE++ behaves differently on two 3D datasets as shown in Table 2. On *BraTS*, ViT-AE++ is competitive to existing contrastive learning-based methods including MoCO-v3 (Chen et al., 2021) and SimSiam (Chen et al., 2020), achieving AUCs of 0.767 *vs.* 0.795 and 0.767 *vs.* 0.771 respectively. This may be caused by overfitting since *BraTS* has only 260 samples in each training fold. On *EGD* which contains two times amount of training samples, ViT-AE++ outperforms the two methods by a large margin, achieving AUCs of 0.846 *vs.* 0.734 and 0.846 *vs.* 0.741 respectively. Notably, the proposed new loss functions consistently improve the representation in the ViT-AE framework. As shown in Table 3, we observe that the auxiliary compound loss and contrastive loss improve vanilla ViT-AE significantly. Especially, to demonstrate the effectiveness of our proposed auxiliary compound loss, we visualize the reconstruction results by different combinations of reconstruction loss functions shown in Figure 3.

Table 2: Comparison of ViT-AE++ and other self-supervised methods.

| Methods | *BraTS* AUC | *EGD* AUC |
|---|---|---|
| MoCO-v3 | **0.795** $\pm$ 0.065 | 0.734 $\pm$ 0.048 |
| SimSiam | 0.771 $\pm$ 0.065 | 0.741 $\pm$ 0.039 |
| Vallina ViT-AE | 0.696 $\pm$ 0.079 | 0.828 $\pm$ 0.036 |
| ViT-AE++(ours) | 0.767 $\pm$ 0.068 | **0.846** $\pm$ 0.034 |

Table 3: Ablation study of the auxiliary loss and contrastive loss on the BraTS dataset.

| Methods | *BraTS* AUC |
|---|---|
| ViT+$\mathcal{L}_{CL}$ | 0.680 $\pm$ 0.091 |
| ViT-AE | 0.696 $\pm$ 0.057 |
| ViT-AE+$\mathcal{L}_{edge}$ | 0.704 $\pm$ 0.065 |
| ViT-AE+$\mathcal{L}_{Per}$ | 0.721 $\pm$ 0.085 |
| ViT-AE+$\mathcal{L}_{edge}$+$\mathcal{L}_{Per}$ | 0.734 $\pm$ 0.088 |
| ViT-AE+$\mathcal{L}_{edge}$+$\mathcal{L}_{Per}$+$\mathcal{L}_{CL}$ | **0.767** $\pm$ 0.069 |

**Does backbone matter?** Our proposed framework used the vision transformer (ViT) as our feature extraction backbone. One might argue that the observed performance improvement is a consequence of using the ViT and not the proposed auxiliary training objectives. In such a scenario, plugging in the ViT in other self-supervised representation learning methods such as MoCOv3 (He et al., 2020) and SimSiam (Chen and He, 2021) should lead to a corresponding increase in representation strength. Tab. 4 shows the results on the *BraTs* and *EGD* datasets. The self-supervised representation features learned using vision transformers perform poorly compared to their ResNet counterpart. We believe this results from overfitting due to a large number of model parameters in ViT. Thus, we conclude that a simple replacement of the feature extractor *does not* guarantee superior performance.

**Analysis of hyperparameters.** We analyze two critical parameters in the proposed framework which affect the representation and training stability.

*Effect of masking ratio $p$.* The masking ratio $p$ determines what percentage of the input volume is masked away. This, in turn, controls the difficulty of the reconstruction task. A higher value of $p$ implies fewer visible patches $X^*$, which makes reconstruction more challenging. On the other hand, a low $p$ can allow the model to extrapolate between visible patches, thus not learning good features. To find a suitable value of $p$, we run experiments

Table 4: Performances of vision transforms trained using different self-supervised representation learning methods.

| Methods | BraTS AUC | EGD AUC |
|---|---|---|
| MoCOv3 | $0.795 \pm 0.057$ | $0.734 \pm 0.048$ |
| SimSiam | $0.771 \pm 0.065$ | $0.741 \pm 0.039$ |
| SimSiam (ViT) | $0.605 \pm 0.088$ | $0.774 \pm \text{red}0.033$ |
| MoCov3 (ViT) | $0.714 \pm 0.069$ | $0.798 \pm 0.029$ |

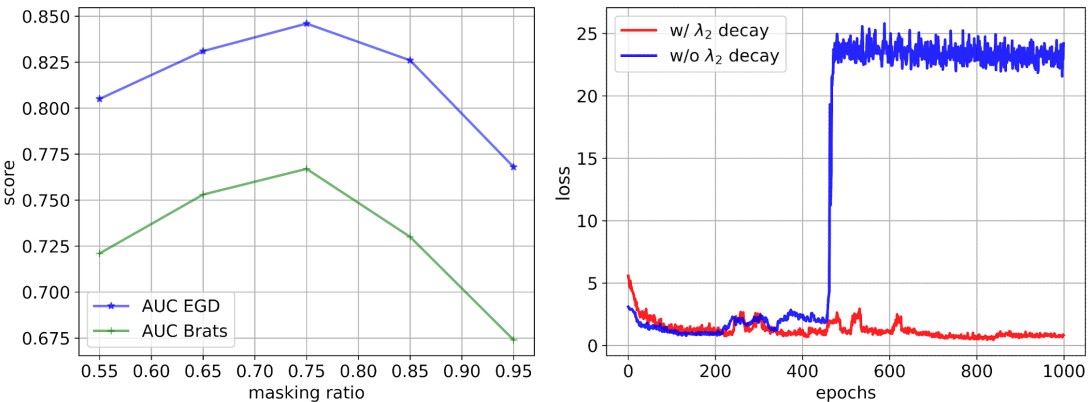

Figure 2: Effects of two key parameters. **Left:** Masking ratio *vs.* AUC on the *BraTS* dataset. The AUC and specificity increase with higher masking ratio, peaking at 0.75, after which they begin to decrease. **Right:** Training loss with Eq. 4 over epochs with and without weight decay of $\lambda_2$ for $\mathcal{L}_{edge}$. Without decay (blue), the loss may explode and the training could not converge; With decay (red), training is stable and converges.

with $p \in [0.55, 0.65, 0.75, 0.85, 0.95]$. The results for the *BraTS* dataset are summarized in Figure 2. We observe that the highest AUC value is obtained at $p = 0.75$. In our experiments, the optimal value was $p = 0.75$ for both datasets.

*Weight decay for edge-based loss.* The edge-based loss weight $\lambda_2$ in Eq. 4 was decreased linearly with number of epochs. We observed that gradually decreasing $\lambda_2$ was essential for the training stability. One possible explanation for the instability is related to the one-to-multiple mapping of the reconstruction task. While the edge map $\mathcal{L}_{edge}$ provides strong additional supervision signals for reconstruction, it is sensitive to noise artifacts which can cause large losses. Fitting to such artifacts destabilizes the training process and hampers the embedding features. To avoid this overfitting, we start the training with $\lambda_2 = 0.01$ and linearly decrease it with each epoch. This allows the network to learn structural information initially and gradually focus more on perceptual similarity. The loss plot is shown in Fig. 2. Without a weight decay of $\lambda_2$, the loss tends to diverge. When using a linear decay, the training is stable and gradually converges.

## 6. Discussion and conclusion

We proposed ViT-AE++ to improve the existing vision transformer-based self-supervised learning approach, which is orthogonal to the existing contrastive learning-based approaches. We introduce a new auxiliary reconstruction loss to the vision transformer autoencoder and extend it with a contrastive loss. We demonstrate that our proposed method is superior to vanilla ViT-AE and competitive to contrastive learning-based methods We hope our work can provide a new perspective for representation learning in medical imaging and advance self-supervised features to the next level. In this work, we focus on learning representation on a single dataset and evaluate it in a downstream task using the same dataset. In future work, we will investigate learning generalizable self-supervised representations immune to common domain shifts (e.g., caused by image acquisition).

## Acknowledgments

We would like to thank the Helmut Horten Foundation for supprting our research. Additionally, Hongwei Bran Li is supported by an Nvidia Academic GPU grant and Forschungskredit (grant No. K-74851-01-01) from the University of Zurich.

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

# Appendix A.

## A.1. Reconstruction results

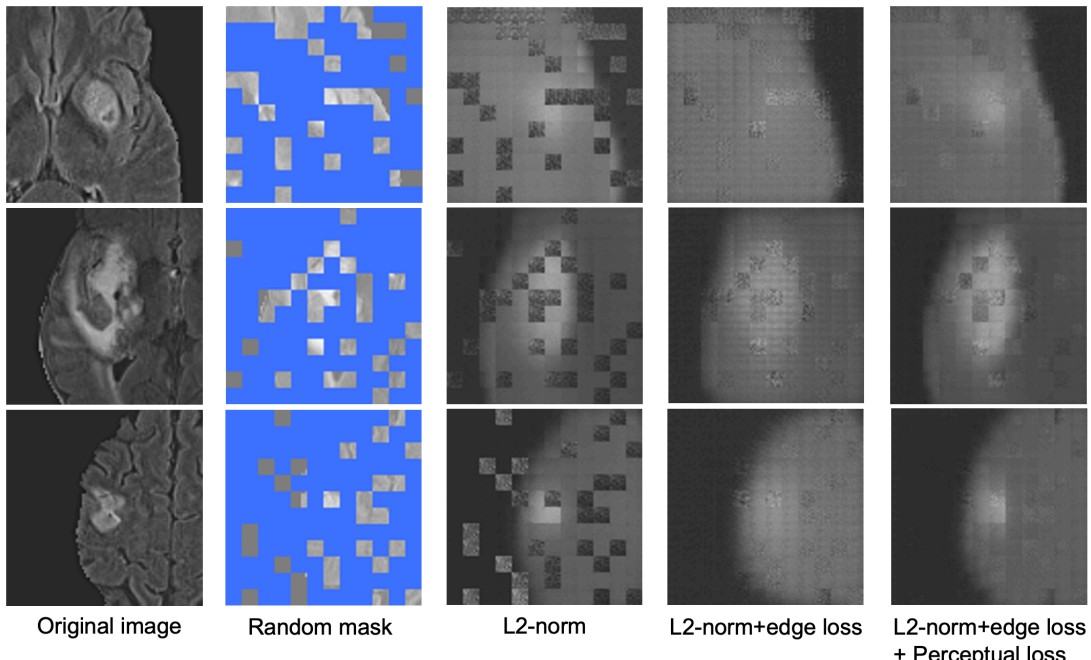

| Original image | Random mask | L2-norm | L2-norm+edge loss | L2-norm+edge loss + Perceptual loss |

Figure 3: Effect of different loss functions for image reconstruction. We observe the progressive improvement of the reconstructed results from a fraction (25%) of the input.

Table 5: Results of two qualitative metrics on image quality of reconstruction results generated by different methods in Fig. 5. The values are computed between the reference image and the reconstructed image.

| Methods | Sample 1 PSNR/SSIM | Sample 2 PSNR/SSIM | Sample 3 PSNR/SSIM |
|---|---|---|---|
| $\mathcal{L}_2$ norm | 21.475/0.396 | 24.064/0.558 | 25.362/0.521 |
| $\mathcal{L}_2+\mathcal{L}_{edge}$ | 24.497/0.503 | 26.751/0.565 | 26.891/0.571 |
| $\mathcal{L}_2+\mathcal{L}_{edge}+\mathcal{L}_{per}$ | **25.339/0.514** | **27.211/0.572** | **27.703/0.586** |

## A.2. 3D Positional Encoding

In our setup, ViT deals with $96 \times 96 \times 96$ cubes. The input volume is divided into patches of $8 \times 8 \times 8$. A special *CLS* token is added to the input sequence (see Fig. 4). This token is used for downstream calculations. Fixed 3D sinusoidal encoding is used to inject positioning information to these patches, including the *CLS* token. There are 1729 input tokens (1728 patch tokens and one *CLS* token). Each axis encodes approximately $\frac{1}{3}$ of the input volume. The framework is flexible to work with different patch sizes.

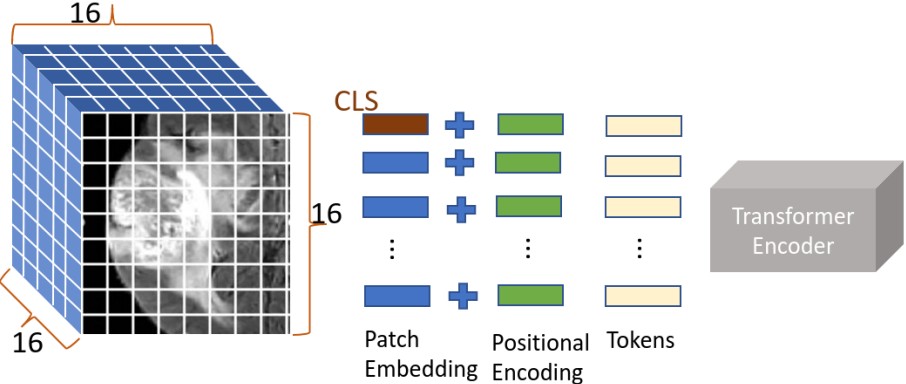

Figure 4: Position encoding representation. The given input volume is divided into a grid of size 16 × 16 × 16. A special CLS token is added along with the patch embeddings. 3D sinusoidal position encoding is added to each input patch (including the special CLS token). The position-aware tokens are fed into the Transformer encoder.

### A.3. Perceptual loss and edge loss

For medical data, a pre-trained VGG-16 (Simonyan and Zisserman, 2015) is used to compute the perceptual similarity between the original volume and its reconstruction. The VGG network is pre-trained on 2D images and can not be directly used with 3D volumes. Hence, we compute loss along 2D slices of the input volume and its reconstruction. The per-slice loss values are averaged to obtain loss for the entire volume. The 3D *Sobel* filter operation is shown in Fig. 5.

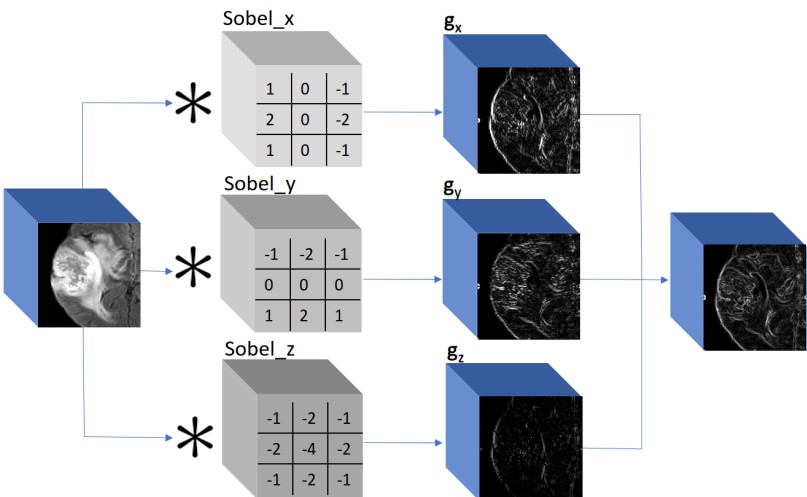

Figure 5: Schematic diagram of 3D Sobel filter. The 3D volume is convolved by 3 × 3 filters. The filter weights are fixed, and each filter computes the gradients in one specific direction. Sobel_x computes the gradients $g_x$ along the Sagittal plane (x-axis). Similarly, $g_y$ and $g_z$ are gradients along the Axial (y-axis) and Coronal (z-axis) planes, respectively. The final edge map is obtained by $\sqrt{g_x^2 + g_y^2 + g_z^2}$.

## A.4. Sample X-ray images.

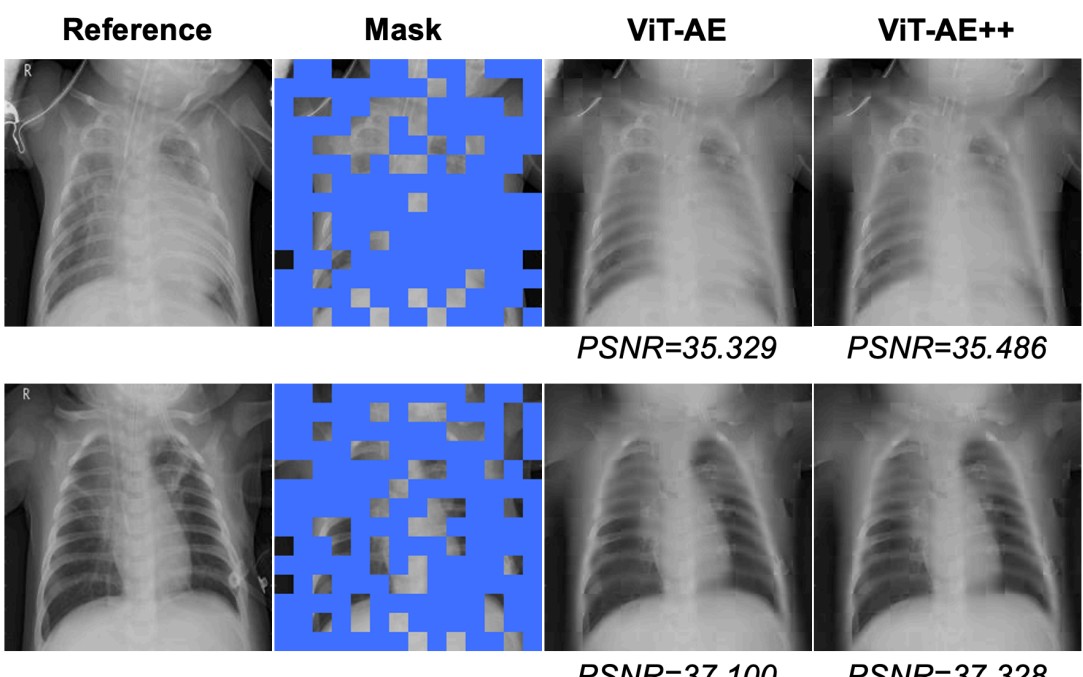

Figure 6: We observed that ViT-AE++ achieved significantly higher PSNR (p-value < 0.0001) than ViT-AE. There is no statistical significance in SSIM.

