# OpenReview forum: "ViT-AE++: Improving Vision Transformer Autoencoder for Self-supervised Medical Image Representations"
_MIDL.io/2023/Conference — MIDL 2023 Poster_

### Official Review · Reviewer_wGMq · 2023-01-29

**Confidence:** 4
**Preliminary Rating:** 4

**Summary:**

This paper presents a new self-supervised method based on ViT-AE for learning 2D/3D medical image representations, by introducing a self-reconstruction task that considers structural dependencies and a contrastive learning task into the vanilla ViT-AE. The proposed method is evaluated on both 2D and 3D image datasets, and the experimental results show its superiority over the vanilla ViT-AE and other contrastive learning methods.

**Strengths:**

The proposed method can be taken as an improvement for the original ViT-AE, it can not only perform the pixel-wise reconstruction task but also consider the structural dependencies in the reconstruction task, which is important for learning medical image representations. Besides, integrated contrastive learning is also a novel way to improve the AE framework. From the experimental results, the proposed strategies can effectively improve the performance of ViT-AE.

**Weaknesses:**

The description of the proposed method needs to be improved further, signs in the formula are not very clear. The experimental setting in ablation study is not sufficient. In addition, there is an error in the analysis of edge-based loss: The edge-based loss weight $\lambda_1$ is the weight of perceptual loss.

**Deanonymize Review:**

no

**Paper Type:**

methodological development

**Questions To Address In The Rebuttal:**

1) What is the meaning of $x_i$ and $\hat{x_i}$ ?
2) In Table 3, what is the performance of ViT-AE++ if only with contrastive loss, and what if only with perceptual loss?
3) Please revise the error in the analysis of edge-based loss weight $\lambda_1$ (should be $\lambda_2$ according to eq.(4)) in the last paragraph of page 7.

---

### Official Review · Reviewer_j2Rz · 2023-02-04

**Confidence:** 5
**Preliminary Rating:** 4
**Recommendation:** Poster

**Summary:**

This paper introduces two loss terms to the MaskAE framework. The compound loss encourages the model to preserve the edge and perceptual consistency between the original and synthesis images. And the constructive learning loss is introduced for a better presentation. Extensive experiments are conducted on natural image benchmarks and medical data sets.

**Strengths:**

1. The paper is well-written and easy to follow.
2. Extensive experimental results on natural image benchmarks and medical data sets are presented.
3. Experimental settings are specified, which facilitates to reproduce the results presented in the paper

**Weaknesses:**

1. The edge consistency loss, perceptual loss, and contrastive loss are well-studied and have been utilized extensively to enhance model representation in various computer vision scenarios. From this perspective, the contribution of the paper is limited.
2. It is not clear from which layers the VGG representations are used to calculate the perceptual loss. If the representation of the early layers  are also included in the perceptual loss calculation, why we need the edge consistency loss? Generally speaking, representation from early layers usually contains such information already. I am wondering what is the performance if only perceptual loss and contrastive loss are incorporated in model training.

**Deanonymize Review:**

no

**Detailed Comments:**

Please refer to the weakness section.

**Paper Type:**

validation/application paper

**Questions To Address In The Rebuttal:**

 It is not clear from which layers the VGG representations are used to calculate the perceptual loss. If the representation of the early layers  are also included in the perceptual loss calculation, why we need the edge consistency loss? Generally speaking, representation from early layers usually contains such information already. I am wondering what is the performance if only perceptual loss and contrastive loss are incorporated in model training.

---

### Official Review · Reviewer_w5bn · 2023-02-05

**Confidence:** 3
**Preliminary Rating:** 4
**Recommendation:** Poster

**Summary:**

Self-supervised learning has attracted increasing attention as it learns data-driven representation from data without annotations. Vision transformer-based autoencoder (ViT-AE) is a recent self-supervised learning technique that employs a patch-masking strategy to learn a meaningful latten space. The manuscript focuses on improving ViT-AD (ViT-AD++) to represent 2D and 3D images adequately. They propose two new loss functions: 1) Aims to improve self-reconstruction; 2) Leverages contrastive loss to optimize the representation from two randomly masked views directly. The method is extensively evaluated on natural and medical images, demonstrating consistent improvements over vanilla ViT-AE and its superiority over other constructive learning approaches.



**Strengths:**

-	They introduce and auxiliary reconstruction task that considers structural dependencies to complement the pixel-wise reconstruction.
-	They unite two paradigms of contrastive learning-based and autoencoder-based methods and enjoy the benefits of both.
-	In extensive experiments, they demonstrate that 2D and 3D ViT-AE++ outperform the vanilla ViT-AE and its superiority over other contrastive learning approaches, setting up a strong baseline for learning self-supervised medical image representations.


**Weaknesses:**

-	The AUCs for the 3D BraTS dataset are similar to the existing contrastive learning-based methods. This may be caused by overfitting since BraTS has only 260 samples in each training fold.
-	The results of the different combinations of reconstruction lost functions are visually disappointing.


**Deanonymize Review:**

no

**Paper Type:**

methodological development

**Questions To Address In The Rebuttal:**

The questions to address in the rebuttal are:
-	Confirm that the low performance for the BraTS dataset is caused by overfitting.
-	Include quantitative measures of the distance between the results and the original images in Figure 2.

---

### Official Review · Reviewer_MJLu · 2023-02-06

**Confidence:** 4
**Preliminary Rating:** 4
**Recommendation:** Oral, Poster

**Summary:**

This work improved the vision transformer-based autoencoder (ViT-AE) by introducing 1) edge loss, 2) perceptual loss, and 3) contrastive loss. Experimental results demonstrated that the proposed ViT-AE++ outperformed the original ViT-AE in 4 2D datasets and 2 other self-supervised methods in two 3D applications. Ablation studies have been conducted to evaluate the importance of each introduced component.

**Strengths:**

1. The manuscript was well-written.
2. The proposed ViT-AE++ showed promising improvement compared to ViT-AE in both 2D and 3D application.
3. Ablation studies have been conducted to evaluate the importance of each introduced component.

**Weaknesses:**

1. The training of proxy and target tasks is not clear. For example, does proxy and target training use the same training set?
2. There are some minor errors in the manuscript (see "Questions To Address In The Rebuttal" for details).

**Deanonymize Review:**

no

**Detailed Comments:**

1. The image reconstruction results (Figure 2) are not particularly relevant to the main topic of this paper. Instead, Appendix A.3. is more interesting and highlights the importance of the integration of different SSL approaches. I would suggest the authors switch Figure 2 and Table 4 and the corresponding contents.
2. Please report both mean and std for the sake of rigorousness.

**Paper Type:**

methodological development

**Questions To Address In The Rebuttal:**

1. Pease elaborate on the details of proxy and target task training. For example, does proxy and target training use the same training set?
2. Page 4, the "Contrastive loss" subsection, "Chen et al., 2020" does not introduce ‘stopping gradient'. Please correct the reference.
3. Figure 3, left panel, what is the score exactly? If it is the test AUC, the results are not consistent with Table 2.

---

### Meta-Review · Area_Chair_D9to · 2023-02-23

**Recommendation:** Accept (Poster)
**Confidence:** 4

**Metareview:**

The rebuttal satisfactorily addressed the reviewers' criticisms.